# An Evaluation of Programmatic Community-Based Chest X-ray Screening for Tuberculosis in Ho Chi Minh City, Vietnam

**DOI:** 10.3390/tropicalmed5040185

**Published:** 2020-12-10

**Authors:** Lan Huu Nguyen, Andrew J. Codlin, Luan Nguyen Quang Vo, Thang Dao, Duc Tran, Rachel J. Forse, Thanh Nguyen Vu, Giang Truong Le, Tuan Luu, Giang Chau Do, Vinh Van Truong, Ha Dang Thi Minh, Hung Huu Nguyen, Jacob Creswell, Maxine Caws, Hoa Binh Nguyen, Nhung Viet Nguyen

**Affiliations:** 1Pham Ngoc Thach Hospital, Ho Chi Minh City 700 000, Vietnam; nguyenhuulan1965@gmail.com (L.H.N.); do_giang68@yahoo.com (G.C.D.); drvinhpnt2020@gmail.com (V.V.T.); hadtm2018@gmail.com (H.D.T.M.); 2Friends for International TB Relief, Ho Chi Minh City 700 000, Vietnam; luan.vo@tbhelp.org (L.N.Q.V.); tmduc93@gmail.com (D.T.); rachel.forse@tbhelp.org (R.J.F.); 3Interactive Research and Development, Singapore 238884, Singapore; 4IRD VN, Ho Chi Minh City 700 000, Vietnam; thang.dao@tbhelp.org; 5Ho Chi Minh City Public Health Association, Ho Chi Minh City 700 000, Vietnam; thanhnguyen246@yahoo.com (T.N.V.); letruonggiang05@gmail.com (G.T.L.); 6Clinton Health Access Initiative Vietnam, Ha Noi 100 000, Vietnam; tluu@clintonhealthaccess.org; 7Ho Chi Minh City Department of Health, Ho Chi Minh City 700 000, Vietnam; bsnguyenhuuhung102@gmail.com; 8Stop TB Partnership, 1218 Geneva, Switzerland; jacobc@stoptb.org; 9Department of Clinical Sciences, Liverpool School of Tropical Medicine, Liverpool L3 5QA, UK; maxine.caws@lstmed.ac.uk; 10Birat Nepal Medical Trust Nepal, Kathmandu 44600, Nepal; 11Viet Nam National Lung Hospital, Ha Noi 100 000, Vietnam; nguyenbinhhoatb@yahoo.com (H.B.N.); vietnhung@yahoo.com (N.V.N.)

**Keywords:** tuberculosis, TB, chest X-ray, active TB case-finding, diagnostic algorithm

## Abstract

Across Asia, a large proportion of people with tuberculosis (TB) do not report symptoms, have mild symptoms or only experience symptoms for a short duration. These individuals may not seek care at health facilities or may be missed by symptom screening, resulting in sustained TB transmission in the community. We evaluated the yields of TB from 114 days of community-based, mobile chest X-ray (CXR) screening. The yields at each step of the TB screening cascade were tabulated and we compared cohorts of participants who reported having a prolonged cough and those reporting no cough or one of short duration. We estimated the marginal yields of TB using different diagnostic algorithms and calculated the relative diagnostic costs and cost per case for each algorithm. A total of 34,529 participants were screened by CXR, detecting 256 people with Xpert-positive TB. Only 50% of those diagnosed with TB were detected among participants reporting a prolonged cough. The study’s screening algorithm detected almost 4 times as much TB as the National TB Program’s standard diagnostic algorithm. Community-based, mobile chest X-ray screening can be a high yielding strategy which is able to identify people with TB who would likely otherwise have been missed by existing health services.

## 1. Introduction

An estimated 170,000 people developed tuberculosis (TB) in Vietnam in 2019, resulting in 11,400 TB-related deaths [1]. Vietnam recently completed its second national TB prevalence survey [2], which documented a 59.9% reduction in smear-positive TB prevalence in the 11 years since the country’s first TB prevalence survey was conducted [3]. However, the new prevalence survey results demonstrated that Vietnam’s total estimated TB burden was higher than previously thought, due to improved survey methods and the use of more sensitive TB diagnostic tests. In light of these findings, the World Health Organization (WHO) revised Vietnam’s official TB burden, and between 2018 [4] and 2019 [1], the National TB Program’s (NTP) official estimated TB treatment coverage declined from 83% to 60%. Low TB treatment coverage presents an immense challenge to achieving the WHO’s End TB Strategy’s 90% TB incidence rate reduction target by 2035 [5], as people with untreated TB continue to transmit the disease.

One of the major barriers to improving TB treatment coverage is that a large proportion of people with TB either do not report symptoms (sub-clinical) or they have mild symptoms and/or have been experiencing symptoms for a short-duration of time (low-grade). These individuals often do not feel compelled to seek out screening and care services and, thus, are frequently missed by health programs. Across Asia, modern TB prevalence surveys have shown that 40–79% of adults with culture-confirmed, pulmonary TB do not report a prolonged cough (≥2 weeks) [6], and a recent meta-analysis indicates that the proportion of people with sub-clinical and low-grade TB may be significantly higher in Asia compared to Africa [7]. Vietnam’s second TB prevalence survey results seem to affirm these findings; 42% of the people with TB detected by the survey did not report a prolonged cough [2]. These individuals were only indicated for diagnostic testing through the use of chest X-ray (CXR) screening. However, WHO guidelines on the use of CXR for TB screening indicate that access to high-quality radiography services can also form a barrier to health-seeking [8]. One root cause is the potentially limited availability of radiography equipment and radiologists. Even if high-quality radiography services are available, their facility-based nature and dependency on opening hours, travel time and associated costs further hinder the integration and use of CXR screening for TB in programmatic settings [9].

A second major barrier to improving TB treatment coverage in Vietnam is the continued reliance on acid fast bacilli (AFB) smear microscopy for diagnosis of TB and the need for smear-negative individuals to undergo a lengthy clinical evaluation process in accordance with NTP guidelines [10]. To address some of these barriers, Vietnam’s NTP recently endorsed projects which pilot the use of CXR screening to both triage symptomatic individuals away from follow-on molecular diagnostic testing with the Xpert MTB/RIF assay (Xpert; Cepheid, Sunnyvale, CA, USA) and to indicate testing for people with mild or no TB symptoms [11]. This screening and testing approach is locally dubbed the ‘Double-X’ or ‘2X’ diagnostic algorithm (X-ray to Xpert).

Several recent active case-finding campaigns across Asia have used CXRs to increase the detection of TB, particularly sub-clinical and low-grade TB. On Palawan island in the Philippines, mobile X-ray trucks were used to screen 25,000 people across various target populations [12]. Between 24.0% and 54.7% of those diagnosed with TB did not report a prolonged cough. In Cambodia, mobile X-rays were used to screen over 21,000 older people (aged ≥ 55 years) in the community [13]. If prolonged cough had been used as the only positive screening criterion, just 31.6% the people diagnosed with TB by the initiative would have been detected. In Pakistan, a fleet of mobile X-ray trucks were used to screen over 127,000 people in the community and also those visiting hospital outpatient departments [14]. Just 45.0% of the people screened by CXR reported a prolonged cough. This initiative did not report its case detection yields disaggregated by symptom status, as only computer-aided reading (CAR) software interpretations of CXR images were used to indicate follow-on diagnostic testing. Several other small-scale studies have also used CXR screening among key populations, including household contacts [15,16] and inmates [17,18], and have reported the detection of high proportions of sub-clinical and low-grade TB.

In Vietnam, due to the cost of human resource capacity-building, laboratory and equipment infrastructure and molecular assay kits, the Ministry of Health requires further evidence on the marginal yield and associated costs to formalize the scale up and transition to the X-ray to Xpert algorithm. We conducted an evaluation to quantify the benefit of using CXR screening and molecular testing in a programmatic, community-based active TB case finding initiative in Ho Chi Minh City, Vietnam.

## 2. Materials and Methods

### 2.1. Study Setting

Between 24 March 2018 and 13 October 2019, Friends for International TB Relief and its partners organized 114 days of CXR screening for TB at 235 unique locations (Figure 1) across seven intervention districts of Ho Chi Minh City, Vietnam (Districts 06, 08, 12, Binh Chanh, Go Vap, Hoc Mon and Tan Binh). These seven districts had a combined population of over 3.4 million people in 2017, the year before this intervention began, and their combined TB notification rate was 141 per 100,000 population. Each district contained one District TB Unit (DTU), which managed TB diagnosis and treatment in collaboration with a network of Commune Health Stations and under the technical supervision of the Provincial TB Program at Pham Ngoc Thach Hospital (PNTH).

These seven districts were selected for inclusion in a two-year, controlled evaluation study comparing the performance of different community healthcare worker (CHW) employment models on the yields, impact and cost of community-based active TB case-finding due to their low case detection and notification rates [19]. In this evaluation study, CHWs screened household contacts and other urban priority groups for symptoms of TB and made referrals for facility-based CXR screening services and follow-on diagnostic testing [20]. Community-based, mobile CXR screening was implemented as a secondary strategy to reduce some of the barriers study participants faced when trying to access facility-based radiography services.

### 2.2. Participant Mobilization

The location for each CXR screening event was selected at least two weeks in advance of implementation. Commune Health Stations, local government offices, schools and churches/pagodas were the most commonly selected sites for screening due to their prominent placement in communities, open spaces for group congregation and availability of water, stable electricity and sanitary facilities. The project’s X-ray truck was frequently moved to a second or third location during the day in an attempt to further reduce geographical access barriers to screening. Community members living in the catchment area of each screening location were sensitized about TB and the availability of upcoming CXR screening services with support from the District- and Commune-level government health staff, civil society organizations (e.g., Women’s Union, Retirement Association, Red Cross, etc.) and the study’s network of CHWs. Participant mobilization was enhanced through the distribution of individualized letters of invitation from each Commune’s People’s Committees, notices on community announcement boards and banners hung in highly visible areas. To maximize TB yields, we targeted urban priority populations for TB, including household contacts, older individuals (≥55 years), people living with HIV or diabetes, people living in socioeconomically disadvantaged areas and others who had TB symptoms. CXR screening events were primarily hosted on Saturdays and Sundays, and occasionally on Fridays, in an attempt to improve access for people who were otherwise obligated during the work week. The events provided screening for additional diseases such as chronic obstructive pulmonary disorder (COPD), asthma, high blood pressure and diabetes to reduce any perceived stigma associated with hosting TB-focused events among the target populations. A VND 10,000 (approximately USD 0.44) cash incentive was provided to all study participants who completed CXR screening, to defray some of the costs of travel to the event.

### 2.3. Screening and Diagnosis of TB

At the screening events, medical student volunteers and CHWs administered a TB symptom questionnaire using a custom-built mHealth app loaded onto Android tablets (Clinton Health Access Initiative/TechUp, Ha Noi, Vietnam), but symptom-screening data were not used to indicate follow-on diagnostic testing. Instead, all participants were referred for CXR screening, whether or not they had symptoms of TB. Digital CXR images were immediately read and classified by a certified radiologist who was instructed to intentionally over-read the CXR images, in line with WHO’s guidelines for CXR classification during TB prevalence surveys [21]. All individuals with an abnormal CXR were guided to provide a good-quality sputum specimen using an instructional aide. Sputum specimens were stored locally until the end of each screening day and were subsequently transported to a nearby laboratory for testing with Xpert.

If an individual was unable to provide a good-quality sputum specimen during the event, a CHW attempted another collection at their home to improve sputum submission rates. People diagnosed with drug-sensitive TB were initiated on treatment at the District TB Units, while people with rifampicin-resistant Xpert results were referred to PNTH for additional testing and drug-resistant TB treatment initiation.

### 2.4. Statistical Analyses

De-identified data were extracted from the study’s mHealth database. The yields at each step of the TB screening cascade [22] were then tabulated, disaggregated by cough duration, age group and gender. Pearson’s χ2 test was used to measure the significance of differences between cohorts of participants who reported having a prolonged cough (≥2 weeks) and those with a cough of short duration (<2 weeks) or no cough at all.

We estimated the marginal yields of bacteriologically-confirmed (Bac[+]) TB for various screening and diagnostic algorithms using a mix of study data (e.g., proportions of participants having a prolonged cough or a cough of any duration) and AFB smear microscopy testing performance characteristics from the literature. Actual Xpert yields were converted into estimated culture yields using an 85% sensitivity rate for Xpert [23] and then AFB smear microscopy yields were estimated using a 43% sensitivity rate [24]. We did not attempt to estimate the yields of TB for participants with symptoms and normal CXR results, as the TB prevalence in this cohort was estimated to be <10 per 100,000 in Vietnam’s second TB prevalence survey [2]. The following six screening and diagnostic algorithms were investigated: (1) cough for ≥2 weeks followed by AFB smear microscopy testing, (2) cough for ≥2 weeks followed by Xpert testing, (3) any cough followed by AFB smear microscopy testing, (4) any cough followed by Xpert testing, (5) CXR abnormal followed by AFB smear microscopy testing and (6) CXR abnormal followed by Xpert testing.

Finally, we estimated the relative diagnostic costs and cost per Bac(+) TB case incurred by the health system for each of the aforementioned screening and diagnostic algorithms. Diagnostic costs were obtained from a Vietnamese health systems costing study [25], including USD 0.18 per AFB smear microscopy test performed and USD 11.12 per Xpert test performed (USD 0.83 for consumables, USD 9.98 for the Xpert cartridge [26] and USD 0.31 for shipping (obtained from the study’s Cepheid shipping documents)). The cost of a provincial-level laboratory technician (USD 216 per month) was also included for the full 19-month project duration. When a diagnostic algorithm resulted in more than an average of 20 sputum tests being conducted per working day, the costs for additional laboratory technicians were included, in line with global TB laboratory standards [27]. We estimated the cost of each CXR image captured and read to be USD 1.53, based on the study’s contract with the X-ray truck rental company (VND 35,000 per image) and used the average interbank exchange rate during the study period to convert into USD (VND 22,949 per 1 USD) [28]. We modeled the number of CXR screens and sputum tests for each algorithm and applied the aforementioned unit costs and added the laboratory technician costs. The cost per Bac(+) case was calculated by dividing the total diagnostic testing costs by each algorithm’s estimated yield. We did not differentiate between the detection of drug-sensitive and -resistant TB; the incremental costs for confirming a drug-resistant TB diagnosis have been reported elsewhere [25].

### 2.5. Ethical Considerations

The Institutional Review Board of PNTH (155/NCKH-PNT) granted scientific and ethical approval for this study. The Provincial People’s Committee of Ho Chi Minh City approved the implementation of the intervention (4699/QD-UBND).

## 3. Results

### 3.1. Yields of TB

Table 1 shows the yields of TB at defined points along the screening cascade. A total of 34,529 participants were screened by CXR, an average of 302 people per day. Over 4700 people had their CXR image classified as abnormal (13.7%) and 2670 people were tested on Xpert (56.5% of eligible). Screening resulted in the detection of 256 people with Xpert-positive TB (9.6% of those tested) translating to an overall TB detection/prevalence rate of 741 per 100,000. Furthermore, 84.4% of participants diagnosed with TB were male and 62.9% were aged 55 years and over.

Just 27.3% of all participants (*n* = 9430) reported having a cough lasting ≥2 weeks. This cohort of participants had a significantly higher CXR abnormality rate (16.1% vs. 12.8%, *p* < 0.001), sputum collection and testing rate (64.3% vs. 52.9%, *p* < 0.001) and Xpert test positivity (13.1% vs. 7.5%, *p* < 0.001) compared to participants with a cough of short duration or no cough at all. Participants with a prolonged cough had a TB detection/prevalence rate of 1357 per 100,000 screened, compared to 510 per 100,000 screened for participants with a cough of short duration or no cough at all. However, only 50% of the total number of people diagnosed with TB by the study (128/256) were detected among participants with a prolonged cough. Among those diagnosed with TB, a higher proportion of people with sub-clinical and low-grade TB was observed among older compared to younger people (52.8% vs. 45.3%) and men compared to women (51.9% vs. 40.0%).

### 3.2. Marginal Yield Analysis and Relative Diagnostic Costing

Table 2 shows the results of the marginal yield analysis and relative diagnostic costing. If the NTP’s standard diagnostic algorithm of testing people with a cough for ≥2 weeks with AFB smear microscopy had been followed, we estimate that just 65 would have been diagnosed with smear-positive TB. This algorithm has the lowest total diagnostic costs at USD 9915 and second lowest cost per Bac(+) case detected of USD 153.12. Identification of cough of any duration, followed by Xpert testing proved to the most expensive algorithm to implement overall at USD 176,131. Although this algorithm is estimated to detect 2.8× more people with TB than the NTP’s standard diagnostic algorithm, it still had a cost per case of USD 978.51. CXR screening followed by Xpert testing detected the most people with TB-almost 4× as many as the NTP’s standard diagnostic algorithm. Since CXRs were used to triage people with symptoms away from Xpert testing and the cost for a CXR is substantially lower than that of an Xpert test in Vietnam, the overall diagnostic costs were cheaper than direct Xpert testing for participants with either a cough of any duration or only those with a cough lasting ≥2 weeks. The X-ray to Xpert algorithm had an overall cost of USD 109,274 and a cost per TB case detected of USD 426.85. This cost per case figure was substantially lower than any other diagnostic algorithm involving Xpert testing.

## 4. Discussion

This evaluation shows that programmatic community-based CXR screening followed by Xpert testing can result in high yields of TB among people who are likely to be missed by existing government health services. We recorded an Xpert-positive TB prevalence which is more than 2.5× the Xpert-positive TB prevalence in the Southern Region of Vietnam [2], indicating that our community sensitization efforts were effective at targeting and mobilizing higher risk individuals. In addition, half of the people diagnosed in this study had sub-clinical or low-grade TB and we estimate that only one quarter of our yield would have been detected if we had followed standard NTP screening and diagnostic guidelines. This finding is in line with a comparison of WHO’s yield estimates for these two diagnostic algorithms during active case finding [29].

The relative diagnostic costing component of this evaluation shows that no matter what screening approach is used, the roll out of Xpert testing services will be significantly more costly than maintaining routine AFB smear microscopy testing. Despite the X-ray to Xpert algorithm having the lowest total costs and cost per TB case of any Xpert-based diagnostic algorithm, it still cost more than 10× the standard NTP diagnostic algorithm, although the cost per case detected was less than 3×. In order to end the TB epidemic, a surge of investment is needed in strategies such as this, so that they can be consistently implemented with enough population coverage and for a sufficient duration of time to impact TB incidence [30]. A recent trial from Vietnam showed that systematic mass diagnostic testing coverage over a 3-year period was associated with a 44% reduction in TB incidence [31]. This trial was able to test between 35–45% of the intervention area’s adult population each year for three years. Due to funding constraints, our study was only able to reach roughly a total 1% of the population living in the intervention districts over a 19-month period.

Our intervention offers a blueprint for an economically viable model that can promote early and increased TB detection, which is essential for breaking the chain of TB transmission in communities and reducing TB incidence. The role which sub-clinical and low-grade TB plays in transmission dynamics is not well understood. However, the literature shows that people with paucibacillary disease, even those with negative nucleic acid amplification tests, can still transmit TB [32,33] and that transmission can occur in the absence of a cough [34]. People with sub-clinical TB may have only transient active infections which self-resolve, or they may be at risk for progression to more serious forms of TB disease [35]. A cohort study from South Africa involving untreated, laboratory-confirmed multidrug-resistant TB (MDR-TB) patients who reported minimal TB symptoms showed that patients with CXR abnormalities were four-times more likely to experience unfavorable outcomes, including death, loss to follow-up or second-line TB treatment initiation, during a 12 month follow-up period [36]. These findings suggest that the people with sub-clinical and low-grade TB who were diagnosed after an abnormal CXR screen by this study are at risk for progressing to more serious forms of TB.

This form of community-based active TB case finding was also a highly person-centered complement to the traditional facility-based referral system, as it directly addressed access barriers. This is evidenced by the higher proportion of women and girls participating in our screening events, even though they have traditionally faced greater access barriers to and longer delays with TB care in Vietnam [37,38]. In our study, for females in particular, it is possible that low sputum collection could explain the observed gender differences in sub-clinical and low-grade TB detection. Studies have shown that simple instructions are able to improve sputum collection among women [39]. We developed and used a visual aide to teach participants how to cough up a good specimen featuring an animated female character, but still recorded a lower sputum collection rate among women. Future community screening should consider additional ways to improve this metric, including by considering where sputum will be collected at the community events and providing a more private place that is shielded from view and inaudible to others for participants to cough up their specimens.

Meanwhile, the effectiveness in our targeting and community mobilization efforts were substantiated by the high case detection among men, particularly in the detection of sub-clinical and low-grade TB. As observed on both prevalence surveys, Vietnam ranks first globally in the male:female ratio (MFR) of people with TB at 5.1 [2,3]. However, the NTP reported an MFR of 2.6 among notified TB patients in 2018 [1], suggesting there are many missed males with TB. Our study has an MFR of 5.4, suggesting that community-based active TB case-finding is critical for reaching persons, especially men, with TB currently missed by the public health system.

There were many programmatic learnings from the implementation of this kind of active TB case-finding. Unlike a TB prevalence survey, participants did not have to be representative of the community in which they live. Thus, the targeting and mobilization of key populations for TB in advance of screening days helped to ensure that CXR throughput and TB yields were sufficiently high to justify this kind of human resource and financial investment. We relied heavily on the health infrastructure of the District TB program; TB Officers from the Commune Health Stations and District TB Units helped to supervise the events from a technical perspective but also helped to mobilize participants in advance of screening days. We also benefited from implementing in a highly structured government system where it was possible to send personalized invitations to all residents of a screening event’s catchment area who were aged over 55 years.

The study used only one X-ray system per event location and we rarely implemented screening events in multiple locations in parallel. We did not purchase an X-ray system for this study as there is a vibrant occupational health screening market in urban Vietnam. There were no missed days due to the lack of X-ray truck rentals, or to damaged X-ray systems; when issues were encountered, a replacement X-ray truck was always available. Although for long-term ACF interventions, procurement of an X-ray system likely makes the most financial sense, this system of commoditized X-ray supply worked extremely well for our short-term study as we were able to rely on private companies that had the scale and technical capacity to meet our needs.

Certified radiologists immediately read and classified all of the CXR images before participants moved to the next station at the screening event. Although the radiologists were instructed to ‘intentionally over-read’ the CXR images, interpretation quality and reader fatigue were real concerns due to the high number of CXRs being read per day. The use of a CAR software to aide CXR interpretation would have been ideal in this high CXR throughput setting. Few CAR software platforms perform as well as experienced radiologists [40,41], and given the availability, quality and cost of radiologists in this setting, it is unlikely that a CAR software would be used to entirely replace human readers. However, a CAR software could be used to reduce workloads by triaging high-confidence normal CXR images, so that the radiologist can focus on providing clinical reports for abnormal images and interpreting the borderline abnormal images. In addition, a CAR software could be used as a live external quality assessment (EQA) tool, such that any abnormal CXR result, either from the radiologist or CAR software, could be used to indicate follow-on Xpert testing. Our study recorded a relatively low sputum collection rate after an abnormal CXR result (56.5%). We believe this was primarily driven by excessive over-reading of CXR images in the first third of our screening events. Using a CAR software as a live EQA tool could also help programs address this challenge by making evidence-based decisions to deprioritize some indicated sputum collection. Prospective feasibility and performance evaluations of CAR software should be conducted in settings where there are few access barriers to radiologists.

One of the most persistent challenges we faced during implementation was ensuring follow-on Xpert testing occurred in a timely fashion. Since a large number of people were screened each weekend, there were an average of 47 sputum specimens waiting every Monday morning to be tested when the laboratories re-opened. Each District TB Unit only has one four-module GeneXpert system, so we had to incentivize laboratory staff to perform tests past normal working hours to ensure a fast turnaround time and minimize sputum quality decay. In urban settings, it may make sense to set up a specialized high-volume Xpert testing facility that can handle the peaks of sputum testing which result from this type of active TB case-finding. Alternately, recent studies from other ACF initiatives in the region have shown that a pooled sputum testing approach could reduce workloads [42].

We estimated the performance of the different algorithms using published literature for the sensitivity of Xpert and AFB smear microscopy. However, these studies were primarily done in passive case-finding settings. Some literature has shown the sensitivity of Xpert during active case-finding to be just 50% [43]. The use of AFB smear microscopy during active case-finding is also highly discouraged due to its suboptimal specificity and common occurrence of false positive results [43]. Since the X-ray to Xpert algorithm is likely to detect people with paucibacillary disease, it is possible that our model over-estimated yields from other screening and diagnostic algorithms. Our study did not test symptomatic individuals with normal CXR results because prevalence survey results indicated that TB yields would be extremely low. We also did not estimate TB yields for this screening cohort when assessing the performance of different diagnostic algorithms for this same reason. However, it is possible that since our study mobilized higher risk populations, that TB yields in this screening cohort could possibly have been higher. We also were unable to assess how diagnosis would be impacted by systematic clinical evaluations, particularly for AFB smear microscopy-based algorithms.

## 5. Conclusions

Programmatic community-based CXR screening can be implemented in urban Vietnam and can achieve high overall yields of TB while detecting a large proportion of people who would likely otherwise have been missed by existing health services. Enhanced case-finding activities such as this are needed in order to improve the coverage of TB testing and treatment so that we can reach the End TB strategy goal of eliminating TB by 2035. However, this strategy is technically more complex and significantly more costly than using routine AFB smear microscopy testing. Thus, additional resources are required in order to scale up and sustain the types of strategies which are necessary to end TB.

## Figures and Tables

**Figure 1 tropicalmed-05-00185-f001:**
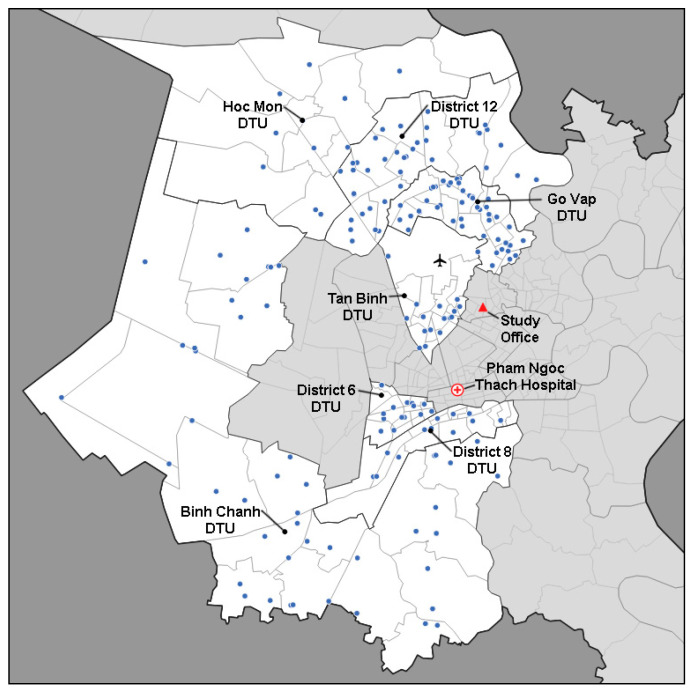
Map mobile chest X-ray (CXR) screening event locations (blue dots) in Ho Chi Minh City, Vietnam.

**Table 1 tropicalmed-05-00185-t001:** Yields of tuberculosis (TB) screening by cough status, age group and gender.

	All Participants	Cough ≥ 2 Weeks	Cough < 2 Weeks and No Cough
All Participants			
Screened by chest X-ray (CXR)	34,529	9430	25,099
CXR abnormal	4722 (13.7%)	1515 (16.1%)	3207 (12.8%)
Tested using the Xpert MTB/RIF assay (Xpert)	2670 (56.5%)	974 (64.3%)	1696 (52.9%)
Xpert(+) TB detected	256 (9.6%)	128 (13.1%)	128 (7.5%)
Xpert(+) TB prevalence rate	741	1357	510
Aged 15–54 Years			
Screened by CXR	12,896	3311	9585
CXR abnormal	1063 (8.2%)	337 (10.2%)	726 (7.6%)
Tested by Xpert	664 (62.5%)	253 (75.1%)	411 (56.6%)
Xpert(+) TB detected	95 (14.3%)	52 (20.6%)	43 (10.5%)
Xpert(+) TB prevalence rate	737	1571	449
Aged ≥ 55 Years			
Screened by CXR	21,633	6119	15,514
CXR abnormal	3659 (16.9%)	1178 (19.3%)	2481 (16.0%)
Tested by Xpert	2006 (54.8%)	721 (61.2%)	1285 (51.8%)
Xpert(+) TB detected	161 (8.0%)	76 (10.5%)	85 (6.6%)
Xpert(+) TB prevalence rate	744	1242	548
Males			
Screened by CXR	14,609	4409	10,200
CXR abnormal	2980 (20.4%)	1023 (23.2%)	1957 (19.2%)
Tested by Xpert	1950 (65.4%)	724 (70.8%)	1226 (62.6%)
Xpert(+) TB detected	216 (11.1%)	104 (14.4%)	112 (9.1%)
Xpert(+) TB prevalence rate	1479	2359	1098
Females			
Screened by CXR	19,920	5021	14,899
CXR abnormal	1742 (8.7%)	492 (9.8%)	1250 (8.4%)
Tested by Xpert	720 (41.3%)	250 (50.8%)	470 (37.6%)
Xpert(+) TB detected	40 (5.6%)	24 (9.6%)	16 (3.4%)
Xpert(+) TB prevalence rate	201	478	107

**Table 2 tropicalmed-05-00185-t002:** Marginal yields of TB from various diagnostic algorithms and their relative diagnostic costs (USD).

	CXR Screens	AFB Tests	Xpert Tests	Estimated TB Yield	Marginal TB Yield	Total Diagnostic Costs	Cost per Bac(+) Case Detected
Cough ≥ 2 weeks followed by acid-fast bacilli smear microscopy (AFB)	0	9430	0	65	-	9915	153.12
Any cough followed by AFB	0	15,101	0	91	+40.6%	10,941	120.16
Cough ≥ 2 weeks followed by Xpert MTB/RIF assay (Xpert)	0	0	9430	128	+97.7%	113,070	883.36
Chest X-ray (CXR) abnormal followed by AFB	34,529	4722	0	130	+100.0%	57,620	444.92
Any cough followed by Xpert	0	0	15,101	180	+178.0%	176,131	978.51
CXR abnormal followed by Xpert	34,529	0	4722	256 *	+295.3%	109,274	426.85

* Actual total yield.

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
