# Peer review of "An Evaluation of Programmatic Community-Based Chest X-ray Screening for Tuberculosis in Ho Chi Minh City, Vietnam"

_tropicalmed, 2020, doi:10.3390/tropicalmed5040185_

Round 1

Reviewer 1 Report

The article is very good but I would like to ask the authors to clarify the details of the cost. How much is the cost for each items should be written in the article.

Line 239; Double-X Algorism means X ray and Xpert?

Author Response

I would like to ask the authors to clarify the details of the cost. How much is the cost for each items should be written in the article.

The paragraph starting on Line 179 has been rewritten to better explain the unit costs, how they were estimated and how they were used.

Line 239; Double-X Algorism means X ray and Xpert?

In Viet Nam, the X-ray to Xpert algorithm has been locally dubbed the “Double-X” or “2X” diagnostic algorithm (see Line 80).  Subsequent mentions of this local name have been replaced in the text to avoid any confusion.

Reviewer 2 Report

Thank you for this description of a community-based TB ACF initiative conducted in Ho Chi Minh city. This study adds to the growing body of literature suggesting that symptom screening as an initial step in TB ACF will miss a substantial number of cases.  

The paper would benefit with further detail as to how the cost benefit and yield analysis was performed (inputs) regarding different screening algorithms, and what this adds/how this differs to existing published analysis. In terms of the algorithms presented, the compelling ACT studies: https://www.nejm.org/doi/full/10.1056/NEJMoa1902129 (citation no.27), did not utilise CXR and instead offered wider testing with Xpert. It would be of interest to include this algorithm in the analysis.

With regards to specific feedback:

  • At lines 102-104: 7 intervention districts are initially mentioned. But subsequent text referrers to 6 districts, please clarify.

  • At lines 161-163, and section at line 205, regarding the methodology of mixing study data and published test performance data in the cost benefit and diagnostic yield analysis: Aside from the smear and xpert sensitivity mentioned, what were the other inputs into this analysis? There is insufficient detail into how this was done. And how does this add to or differ from the similar analysis provided in the WHO guidance? https://apps.who.int/iris/bitstream/handle/10665/181164/9789241549172_eng.pdf?sequence=1

  • At lines 318: were specimens appropriately stored? Previous work has demonstrated that viability of sputum samples for TB testing can be maintained for considerable time if stored at appropriate temperatures.

  • At line 191 and table 1: only 56% of those with an abnormal CXR had a sputum tested with Xpert, this seems to be a major drop off in the diagnostic cascade. Further discussion into how this could be improved would be beneficial e.g. role of sputum induction? etc

Author Response

The paper would benefit with further detail as to how the cost benefit and yield analysis was performed (inputs) regarding different screening algorithms, and what this adds/how this differs to existing published analysis.

The paragraph starting on Line 179 has been rewritten to better explain the unit costs, how they were estimated and how they were used.

In terms of the algorithms presented, the compelling ACT studies: https://www.nejm.org/doi/full/10.1056/NEJMoa1902129 (citation no.27), did not utilise CXR and instead offered wider testing with Xpert. It would be of interest to include this algorithm in the analysis.

In the ACT-3 study, sputum specimens were collected from any community member who could spontaneously cough up a specimen.  Only 5.4% of participants reported a cough on the day of testing.  Unfortunately, we could not test everyone in the study with Xpert due to budget constraints, and so our study data can replicate their study design.

At lines 102-104: 7 intervention districts are initially mentioned. But subsequent text referrers to 6 districts, please clarify.

This was an oversight in the first manuscript.  We have updated the text to always show seven intervention districts.

At lines 161-163, and section at line 205, regarding the methodology of mixing study data and published test performance data in the cost benefit and diagnostic yield analysis: Aside from the smear and xpert sensitivity mentioned, what were the other inputs into this analysis? There is insufficient detail into how this was done.

The paragraph starting on Line 166 was modified to clarify how the different diagnostic yields were estimated.  We used our actual project data to determine the proportion of participants who would’ve screened positive using the different screening tests, and then used the actual Xpert yields within that screening cohort to estimate AFB smear microscopy yields based on the literature.  Similarly, the paragraph starting on line 173 was edited to better capture which unit costs were used and how they were obtained.

And how does this add to or differ from the similar analysis provided in the WHO guidance? https://apps.who.int/iris/bitstream/handle/10665/181164/9789241549172_eng.pdf?sequence=1

In terms of costs, The WHO operational guidelines do not provide detailed estimates of diagnostic costs which are linked to actual or estimated yields from screening.  In fact they state ‘However, the availability, cost and feasibility of tests may vary considerably in different parts of the health-care system.’ They model different approaches based on published studies. Our evaluation estimated the costs of different diagnostic algorithms for a screening program of this scale/size in Viet Nam using country-derived data, a setting with a medium TB burden.  Since this was not a full health systems costing exercise, the costing component was meant to show the relative diagnostics and costs per cases, as even with relatively minimal inputs in this model, there are large differences in costs.  This fits into a global narrative about the need for additional investment in TB, particularly highly sensitive screening approaches to improve TB treatment coverage. Table 6 in the WHO document estimates that symptoms screening followed by SSM would detect 214 B+ cases while a CXR followed by Xpert would detect 902 B+ cases – which is roughly in line with our findings of about 4x the yield (text added in Lines 256-257).

At lines 318: were specimens appropriately stored? Previous work has demonstrated that viability of sputum samples for TB testing can be maintained for considerable time if stored at appropriate temperatures.

At the screening events, sputum specimens were stored in cooler containers, out of direct sun light.  After transportation to the laboratory, specimens were stored in refrigerators until testing.  We agree that with these storage conditions and the use of a molecular assay, the risk of yield loss due to minor testing delays is minimal.  We have removed this line of the paragraph.

At line 191 and table 1: only 56% of those with an abnormal CXR had a sputum tested with Xpert, this seems to be a major drop off in the diagnostic cascade. Further discussion into how this could be improved would be beneficial e.g. role of sputum induction? Etc

We believe the low overall sputum collection rate was caused by excessive over-reading early in our early campaigns.  In our last intensive district-level campaign, we recorded a sputum testing rate of >80%.  We have added sentences in Lines 332-336 to discuss the potential role a CAR software could play in helping identify and respond to excessive over-reading.

Reviewer 3 Report

In general

 the authors addressed the CXR for early detect the PTB

 The PTB was confirmed by the method of X-pert . The conclusion was the CXR a critical role  for identify the PTB.

major comment 

 1 the author should tell us the criteria of the CXR finding indicate the PTB

  such as RUL infiltratin? cavitation?

 2 the sensitivity and  specificity , the PPV , the NPV ?

      X-pert    +    X-pert _

CXR +

CXR -

3 the interobservers interpretation? 

minor comment 

  the underling diseases  of these study cohort ?

Author Response

1 the author should tell us the criteria of the CXR finding indicate the PTB, such as RUL infiltratin? cavitation?

Lines 148-150 indicate that CXR images were interpreted in line the WHO’s TB prevalence survey guidelines.  A citation for these guidelines has been provided.  An abnormal chest X-ray means any lung (including pleura) abnormality, including opacities, cavitation, fibrosis, pleural effusion, calcification, and any unexplained or suspicious shadow.  Congenital abnormalities, bony abnormalities (e.g. fractures), and heart-related abnormalities (e.g. increased heart size) were excluded.

2 the sensitivity and specificity, the PPV, the NPV?

We agree that this would be ideal – but in our study, participants who had a CXR normal result were not indicated for Xpert testing.  Thus, we are unable to populate cells C and D in the proposed 2x2 contingency table and cannot calculate sensitivity (a/a+c), specificity (d/b+d), or NPV (d/c+d).  It would be possible to calculate PPV (a/a+b), but in isolation this metric provides little additional information over the percentage yield of Xpert-positive TB figures in Table 1.

3 the interobservers interpretation?

Unfortunately we did not assess inter-reader variability in this study, as it was a retrospective evaluation of programmatic active case finding and we only had one person read each CXR image.  In the discussion paragraph starting on line 321, we discuss the issue of interpretation quality and reader fatigue using a computer-aided reading (CAR) software.

the underling diseases of these study cohort?

This study did not systematically capture data about the prevalence of other diseases for all participants.  Viet Nam is a low HIV prevalence setting (just about 3% of all TB patients are living with HIV).  While other disease screening services were often offered at the study’s CXR screening events, these services were usually offered to specific sub-populations, rather than all participants, and outcomes of non-TB referrals into the government health system were not tracked by the study.

Round 2

Reviewer 2 Report

Thank you for the revisions and addressing the queries raised

Reviewer 3 Report

the author answer my suggestion the reference 2

the paper have merit for publication